# A Doxorubicin-Glucuronide Prodrug Released from Nanogels Activated by High-Intensity Focused Ultrasound Liberated β-Glucuronidase

**DOI:** 10.3390/pharmaceutics12060536

**Published:** 2020-06-10

**Authors:** Helena C. Besse, Yinan Chen, Hans W. Scheeren, Josbert M. Metselaar, Twan Lammers, Chrit T. W. Moonen, Wim E. Hennink, Roel Deckers

**Affiliations:** 1Division of Imaging and Oncology, University Medical Center Utrecht, 3584 CX Utrecht, The Netherlands; h.c.besse@umcutrecht.nl (H.C.B.); c.moonen@umcutrecht.nl (C.T.W.M.); 2Department of Pharmaceutics, Utrecht Institute for Pharmaceutical Sciences, Utrecht University, 3584 CG Utrecht, The Netherlands; yinanchen.0406@gmail.com (Y.C.); tlammers@ukaachen.de (T.L.); w.e.hennink@uu.nl (W.E.H.); 3Cluster for Molecular Chemistry, Radboud University, 6525 XZ Nijmegen, The Netherlands; J.Scheeren@science.ru.nl; 4Department of Nanomedicine and Theranostics, Institute for Experimental Molecular Imaging, RWTH Aachen University Clinic, 52074 Aachen, Germany; bart@enceladus.nl; 5Department of Targeted Therapeutics, MIRA Institute for Biomedical Engineering and Technical Medicine, University of Twente, 7500 AE Enschede, The Netherlands

**Keywords:** nanogel, prodrug, high-intensity focused ultrasound, local drug delivery, enzyme prodrug therapy

## Abstract

The poor pharmacokinetics and selectivity of low-molecular-weight anticancer drugs contribute to the relatively low effectiveness of chemotherapy treatments. To improve the pharmacokinetics and selectivity of these treatments, the combination of a doxorubicin-glucuronide prodrug (DOX-propGA3) nanogel formulation and the liberation of endogenous β-glucuronidase from cells exposed to high-intensity focused ultrasound (HIFU) were investigated in vitro. First, a DOX-propGA3-polymer was synthesized. Subsequently, DOX-propGA3-nanogels were formed from this polymer dissolved in water using inverse mini-emulsion photopolymerization. In the presence of bovine β-glucuronidase, the DOX-propGA3 in the nanogels was quantitatively converted into the chemotherapeutic drug doxorubicin. Exposure of cells to HIFU efficiently induced liberation of endogenous β-glucuronidase, which in turn converted the prodrug released from the DOX-propGA3-nanogels into doxorubicin. β-glucuronidase liberated from cells exposed to HIFU increased the cytotoxicity of DOX-propGA3-nanogels to a similar extend as bovine β-glucuronidase, whereas in the absence of either bovine β-glucuronidase or β-glucuronidase liberated from cells exposed to HIFU, the DOX-propGA3-nanogels hardly showed cytotoxicity. Overall, DOX-propGA3-nanogels systems might help to further improve the outcome of HIFU-related anticancer therapy.

## 1. Introduction

Chemotherapy is one of the most commonly used treatment modalities in cancer, either as a monotherapy or in combination with another treatment modalities, such as radiotherapy and surgery [1]. The agents used in chemotherapy treatment are often not tumor-cell-specific. Hence, these agents also cause damage to normal tissue, which could ultimately cause severe dose-limiting side effects and reduce the efficacy of chemotherapy treatment [2,3,4].

These chemotherapy-related side effects can potentially be reduced by using prodrug therapy. Prodrugs are noncytotoxic drug precursors that ideally are only activated in the tumor microenvironment into the pharmacologically active cytotoxic drug [5,6]. The activation of the prodrug can be caused by many triggers, like hypoxia, radiation, pH, tumor-specific antigens, and enzymes [6].

In cancer treatment most prodrugs are activated by enzymes, i.e., enzyme prodrug therapy [7,8]. A previously synthesized prodrug for enzyme prodrug therapy is DOX-GA3 [9], which is converted into the cytotoxic agent doxorubicin (DOX) by the enzyme β-glucuronidase (β-gus) [9,10]. β-Gus is a lysosomal enzyme that is limitedly present in the blood [11] and is only extracellularly present in necrotic tumor microenvironment [12,13]. Therefore, there is only conversion of DOX-GA3 into DOX in the necrotic tumor [9]. As a single treatment, DOX-GA3 is 12 times less cytotoxic than DOX, in vitro [10]. The low cytotoxicity of DOX-GA3 is mainly caused by the fact that DOX-GA3 is hydrophilic and therefore not able to cross the cell membrane [14], whereas treatment with DOX-GA3 in combination with β-gus results in vitro has similar efficacy to DOX treatment [9,10]. In addition, in vitro the efficacy of DOX-GA3 is higher than DOX, since DOX-GA3 has a larger maximum tolerated injected dose [9,10]. The therapeutic effectiveness of DOX-GA3, however, can be further increased by increasing its circulation half-life [15].

To improve the pharmacokinetics of DOX-GA3, this prodrug can be loaded into a drug delivery system such as liposomes and nanogels [16,17,18,19,20,21]. Nanogels are nano-sized hydrogel particles consisting of crosslinked hydrophilic polymer chains that can be physically loaded with drugs and biotherapeutics or chemically conjugated with pharmacologically active agents [22]. As shown before, nanogels are able to improve the half-life of small drugs in the circulation [23,24,25]. In addition, nanogels passively target the tumor by the loosely vascular aligning and the lack of lymphatic drainage in tumors, also known as the enhanced permeability and retention (EPR) effect [26,27]. Therefore, nanogels are promising delivery systems for small molecular prodrugs.

Besides the pharmacokinetics, the site selective activation of the prodrug is also an important factor for effective prodrug therapy [28,29]. To achieve effective prodrug therapy treatment, the enzymes that are able to convert the inactive prodrug into its active constituent should be highly expressed in the tumor [30]. Since the β-gus concentrations are only sufficient in large necrotic tumors, in small tumors the efficacy of doxorubicin-glucuronide prodrugs is hampered [31]. Many strategies have been investigated to increase the β-gus concentration available for prodrug conversion in the tumor, like transfecting tumor cells with the gene encoding for β-gus (gene-directed enzyme prodrug therapy (GDEPT)) [32,33] and administration of antibody-enzyme conjugates (antibody-directed enzyme prodrug therapy (ADEPT)) [34]. Currently, these therapies are not used in the clinic, due to insertional mutagenesis in GDEPT, and costs and immunogenicity of ADEPT constructs [30,33,35]. Recently, the concept of ultrasound-directed enzyme prodrug therapy (UDEPT) was introduced by Besse et al. [36]. In this concept, endogenous β-gus is liberated from tumor cells by exposing them to high-intensity focused ultrasound (HIFU). Subsequently, the liberated β-gus from the cells is able to convert the prodrug into the cytotoxic agent. Since HIFU is a local and noninvasive technique [37], this enables the possibility of increasing the β-gus concentration available for prodrug conversion locally in the tumor by a noninvasive treatment, without damaging the normal tissue.

As mentioned, both the pharmacokinetics and tumor side selective activation of the prodrug are important factors for effective prodrug therapy treatment [28,29]. Here, we investigated the combination of prodrug-nanogel formulation and UDEPT to address the shortcomings of small molecular prodrugs and increase the enzyme concentration available for prodrug conversion, in vitro. To this end, doxorubicin-glucuronide prodrug (DOX-propGA3) (structure shown in Figure 1A) was coupled to the polymer hydroxyethyl methacrylamide-oligoglycolates-derivatized poly(hydroxyethyl methacrylamide-co-*N*-(2-azidoethyl)methacrylamide (p(HEMAm-co-AzEMAm)-Gly-HEMAm) via click chemistry (DOX-propGA3-polymer). The formed conjugate was further used for the preparation of nanogels (DOX-propGA3-nanogels), Figure 1B. Subsequently, the conversion of DOX from the DOX-propGA3-nanogel in the presence of bovine β-gus was investigated. Finally, it was confirmed in vitro that β-gus liberated from cells exposed to HIFU was able to increase the cytotoxicity of DOX-propGA3-nanogels.

## 2. Materials and Methods

### 2.1. Materials

DOX.HCl was purchased from Guanyu bio-technology Co., LTD (Xi’an, China). Bovine β-gus (G0376, Type B-3, 2000 units/mg solid) and 4-methylumbelliferyl β-d-glucuronide (M9130, 4-MUG) were purchased from Sigma-Aldrich (Zwijndrecht, The Netherlands). Irgacure 2959 was obtained from Ciba Specialty Chemicals Inc. (Bazel, Switzerland). ABIL EM 90 was provided from Evonik Industries AG (Essen, Germany). Acetonitrile (ACN), dichloromethane (DCM), dimethylformamide (DMF), ethyl acetate, methanol, hexane, and dimethyl sulfoxide (DMSO) were obtained from Biosolve (Valkenswaard, The Netherlands). RPMI 1640 (R8758) and fetal bovine serum (FBS) were purchased from ThermoFisher (Bleiswijk, The Netherlands). CellTiter 96^®^ AQueous One Solution Cell Proliferation Assay (G3580, MTS reagent) was obtained by Promega (Leiden, The Netherlands). All other chemicals and reagents were obtained from Sigma-Aldrich (Zwijndrecht, The Netherlands).

### 2.2. Cell Culture

Mouse mamma carcinoma 4T1 cells (ATCC, ATCC CRL-2539, Rockville, MD, USA) were cultured in RPMI 1640 supplemented with 10% FBS at a temperature of 37 °C in a humidified atmosphere containing 5% CO_2_. Cells were regularly tested mycoplasma negative.

### 2.3. Synthesis of DOX-propGA3-Polymer

DOX-propGA3-polymer conjugate was synthesized as shown in Figure 1A. First, the synthesis of the polymer p(HEMAm-co-AzEMAm)-Gly-HEMAm (further referred to as ‘polymer’) (20 mol% AzEMAm, degree of substitution 10, M_n_ 15 kDa, PDI 3.0) [38] and DOX-propGA3 [39] was performed as previously described. The polymer (100 mg) and DOX-propGA3 (8.9 mg, 0.1 mol prodrug/mol azide) were dissolved in 1.8 mL DMF. Then, a mixture of 1.45 mg CuSO_4_ (1 eq to DOX-propGA3) and 1.8 mg sodium ascorbate (1 eq to DOX-propGA3) dissolved in 200 μL ammonium acetate buffer (100 mM, pH 5) was added. The resulting solution was stirred at room temperature for 24 h under a nitrogen atmosphere. Next, the obtained product was purified by three times precipitation in diethyl ether and dissolving in methanol. Subsequently, the precipitate was dissolved in water and dialyzed (membrane cut-off 3.5 kDa) against ammonium acetate buffer (20 mM, pH 5, containing 10 mM EDTA) for 2 days, followed by dialysis against water for 24 h. The ammonium acetate buffer and water were changed at least six times. Ratios between ammonium acetate buffer and sample and between water and sample were larger than 500. Finally, DOX-propGA3-polymer conjugate was recovered by freeze drying.

### 2.4. Characterization of the DOX-propGA3-Polymer Conjugate

The synthesized DOX-propGA3-polymer conjugate was analyzed by gel permeation chromatography (GPC) using a Waters System (Waters Associates Inc., Milford, MA, USA) with refractive index (RI) and UV detection using two PLgel 5 μm MIXED-D columns (Agilent, Pal Alto, CA, USA) and DMF containing 10 mM LiCl as eluent, with an injection volume of 100 μL, and flow rate of 1 mL/min at a temperature of 60 °C. UV detection of DOX was performed at 480 nm.

The conjugation efficacy of DOX-propGA3-polymer conjugate was determined at a concentration of 0.5 mg/mL in phosphate-buffered saline (PBS). Calibration was done using DOX (10 to 100 µg/mL in PBS). DOX concentration was determined by ultraviolet-visible (UV-vis) spectrophotometry (BMG Labtech, Offenburg, Germany) at an absorbance of 480 nm. The conjugation efficiency and loading capacity were calculated according to Equations (1) and (2), respectively.
(1)$$conjugation\;efficiency=\frac{\mathrm{amount}\;\mathrm{of}\;\mathrm{DOX}-\mathrm{propGA}3\mathrm{conjugated}\;\mathrm{to}\;\mathrm{polymer}}{\mathrm{amount}\;\mathrm{of}\;\mathrm{DOX}-\mathrm{propGA}3\;\mathrm{feed}}\times 100\%$$
(2)$$loading\;capacity=\frac{\mathrm{amount}\;\mathrm{of}\;\mathrm{DOX}-\mathrm{propGA}3\;\mathrm{conjugated}\;\mathrm{to}\;\mathrm{polymer}}{\mathrm{amount}\;\mathrm{of}\;\mathrm{DOX}-\mathrm{propGA}3-\mathrm{polymer}\;\mathrm{conjugate}}\times 100\%$$

### 2.5. Preparation of DOX-propGA3-Nanogels

DOX-propGA3-nanogels were prepared by inverse mini-emulsion photo polymerization as previously described [40], Figure 1B. Briefly, 37.5 mg DOX-propGA3-polymer was dissolved in 412.5 μL DMSO, and subsequently 150 μL Irgacure 2959 (10 mg/mL in water) was added. This mixture was added to 5 mL mineral oil (containing 10% *v*/*v* ABIL EM 90) and thoroughly vortexed. The primary emulsion was ultra-sonicated (Bandelin Sonopuls, pulse on/off 0.5 s, and amplitude 10%) for 15 min and irradiated under UV (60% amplitude, 940 mW/cm^2^, 300–650 nm, Bluepoint UVC source, Hönle UV technology, Gräfelfing, Germany) for 15 min. Subsequently, the mineral oil, surfactant, and DMSO were removed by washing the formed DOX-propGA3-nanogels once with acetone (40 mL) and four times with acetone/hexane (40 mL, 1:1, *v*/*v*). Finally, the DOX-propGA3-nanogels were recovered by re-dispersion in water and freeze drying.

The size of DOX-propGA3-nanogels was measured by dynamic light scattering (DLS) on an ALV CGS-3 system (Malvern Instruments, Malvern, UK) with a JDS Uniphase 22 mW He-Ne laser operating at 632.8 nm, an optical fiber-based detector, digital LV/LSE-5003 correlator at 25 °C, expressed on intensity. The ζ potential of DOX-propGA3-nanogels was measured with Malvern Zetasizer Nano-Z (Malvern, UK) at 25 °C. Measurements were performed in 20 mM HEPES buffer (pH 7.4) at a DOX-propGA3-nanogel concentration of 0.5 mg/mL.

### 2.6. Prodrug Conversion

Both DOX-propGA3-polymer and DOX-propGA3-nanogels were dispersed in phosphate buffered saline (PBS, pH 7.4, containing 0.049 M NaH_2_PO_4_, 0.099 M Na_2_HPO_4_ and 0.006 M NaCl) containing 0.1% (*w*/*v*) bovine serum albumin (BSA) [39] to a final concentration of 100 μg/mL. This corresponds to a concentration of 7 µg/mL DOX. Next, 50 µl of a stock solution of bovine β-gus (2 mg/mL in PBS) was added to yield a final enzyme activity of 100 units/mL in a total volume of 2 mL. As a negative control, the DOX-propGA3-polymer and DOX-propGA3-nanogels were dispersed in the same buffer without bovine β-gus. Samples were incubated in a water bath at 37 °C. After incubation times ranging from of 0 to 48 h, 200 μL samples were taken at different time points and centrifuged (20,000× *g* for 60 min) at 4 °C. Subsequently, DOX concentration in the supernatant was determined using UPLC analysis (Waters ACQUITY UPLC system (Waters Associates Inc., Milford, MA, USA)) using an Acquity BEH C18 column 1.7 μm (2.1 × 50 mm); eluent A and B were potassium phosphate buffer (20 mM, pH 3)/acetonitrile (95/5, *v*/*v*) and 100% ACN, respectively. The injection volume was 5 μL, and fluorescence was detected at a wavelength of 560 nm (excitation wavelength of 480 nm). After an isocratic flow of 75% eluent A for 1 min, a gradient was run from 75 to 60% eluent A in 3 min with a flow rate of 0.5 mL/min. The retention time of DOX was 0.78 min. The calibration curve of DOX was linear between 0.01 and 10 μg/mL. Finally, chromatograms were analyzed by Empower Software, Version 1154.

### 2.7. Induction of β-Gus Liberated from 4T1 Cells by HIFU

Endogenous β-gus was liberated from cells by exposing them to HIFU by an in-house developed HIFU system. HIFU was performed by a single-element transducer (external radius of aperture 120 mm, focal length 80 mm and focal point 1 × 1 × 3 mm^3^ (at −3dB)). Sine-shaped waves were generated by an AG Series Amplifier (AG 1006, T&C Power Conversion Inc. Rochester, NY, USA) at a frequency of 1.3 MHz, a pulse repetition time of 50 ms, a duty cycle of 1% (corresponding to 650 cycles per pulse), and a peak negative pressure of 41 MPa; a schematic representation of the setup is present in Appendix A. Acoustic pressures in the focal point were measured as a function of input voltage using a fiber optic hydrophone in a tank filled with degassed water, see [41] for details. Cells (2 × 10^6^ cells in 170 µl PBS) in a PCR tube (200 µl, Bio rad, California, CA, USA) were exposed to HIFU by positioning this tube in the focus of the HIFU beam for 10 min. Immediately after exposure of the cells to HIFU, samples were placed on ice and either analyzed by microscopy or centrifuged at 16,000× *g* for 15 min at 4 °C. The supernatant after centrifugation was further used to measure the β-gus activity, conversion of DOX-propGA3-nanogels into DOX, and cytotoxicity in combination with DOX-propGA3-polymer and DOX-propGA3-nanogels, as described below.

### 2.8. Microscopy of Cells Exposed to HIFU

Samples of 10 µl from cells exposed to HIFU and untreated cells (negative control) were taken and added to 240 µl cell culture medium in an ibidi chamber of 1µ-Slide 8 Well ibiTreat (Ibidi GmbH, Munich, Germany). Subsequently, samples were 1 h incubated under normal culturing conditions, to allow attachment of the cells to the plate. Finally, samples were imaged by inverted microscopy (ULWCD 0.30, Olympus CK2, Tokyo, Japan) with a digital camera (Moticam 5-5.0 MP, Hong Kong, China) using a 10× objective.

### 2.9. Determination of the β-Gus Activity

β-Gus activity in the supernatant of cells exposed to HIFU and untreated cells (negative control) was measured by a MUG assay adapted from Jefferson et al. [42]. Briefly, 20 µl sample was added to 180 µl 4-methylumbelliferyl β-d-glucuronide solution (1 mg/mL in 0.1 M sodium acetate (pH 4.5)) and incubated for 1 h in a water bath of 37 °C. Subsequently, 950 µl of 0.2 M sodium carbonate (i.e., stopping buffer) was added to 50 µl of all samples. Finally, the fluorescence intensity was measured using a spectrofluorometer (Jasco FP8300, Tokyo, Japan), excitation of 380 nm, and emission of 454 ± 5 nm. The enzyme activity was calculated based on the enzyme activity of commercial bovine β-gus.

### 2.10. Conversion of DOX-propGA3-Nanogels into DOX by β-Gus Liberated from HIFU Treated Cells

Freeze-dried DOX-propGA3-nanogels were dispersed in 0.95 mL PBS containing 0.1% (*w*/*v*) BSA at a DOX concentration of 5 μg/mL. Next, 50 μL of the supernatant of cells exposed to HIFU was added and mixed carefully. After incubation in a water bath at 37 °C for 48 h, the solution was analyzed for DOX concentration by UPLC as described in section “prodrug conversion”.

### 2.11. In Vitro Cytotoxicity

In a 96 well plate, 4T1 cells were seeded at a density of 2,500 cells/well. After 24 h, the cell culture medium was removed and 200 μL of DOX, DOX-propGA3, DOX-propGA3-polymer, and DOX-propGA3-nanogels, in cell culture medium with PBS (10 μL in 190 μL cell culture medium); bovine β-gus (50 μg/mL, enzyme activity of 100 units/mL in cell culture medium); or supernatant of 4T1 cells exposed to HIFU (10 μL in 190 μL cell culture medium) was added to the wells at equivalent DOX concentrations ranging from 2 to 100,000 nM. After 24 h incubation, cells were washed three times with 200 μL PBS and 100 μL fresh cell culture medium was added. Subsequently, MTS assay was performed according to manufacturer’s protocol. Briefly, 20 μL of MTS reagent was added to each well and incubated for 3 h under normal culturing conditions. Finally, the optical density of the different samples was recorded by an EZ Read 400 microplate reader (Biochrom Ltd., Cambridge, UK) at an absorbance of 492 nm; an absorbance of 690 nm was used as background.

### 2.12. Statistical Analysis

All data is presented as mean, with error bars representing the standard deviation of at least three independent experiments. To determine differences in cytotoxicity, a two-tailed student’s t-test was used to determine significance between the IC_50_ value of the different groups. Significant differences were considered as *p* < 0.05.

## 3. Results and Discussion

### 3.1. Synthesis of DOX-propGA3-Polymer Conjugate and DOX-propGA3-Nanogels

P(HEMAm-co-AzEMAm) was synthesized by free radical polymerization using HEMAm and AzEMAm as monomers and ABCPA as initiator as described in detail in our previous publication [38]. The characteristics and ^1^H-NMR spectrum of the obtained polymer are given in Appendix A (from ref. [38]). In the next step, the obtained p(HEMAm-co-AzEMAm) was further modified with HEMAm-Gly (a polymerizable group) to yield p(HEMAm-co-AzEMAm)-Gly-HEMAm [38].

The DOX-propGA3-polymer was synthesized from DOX-propGA3 prodrug, as shown in Figure 1A. The conjugation of DOX-propGA3 to the p(HEMAm-co-AzEMAm)-Gly-HEMAm was performed by Cu(I)-catalyzed azide-alkyne cycloaddition (CuAAC). In this conjugation, step sodium ascorbate was added as reducing agent to generate Cu(I) from the Cu(II) salt (CuSO_4_) instead of directly adding active Cu(I) to the reaction since conjugation does not occur by using active Cu(I) in the reaction of this sterically hindered doxorubicin molecule with the bulky polymer [39]. After the reaction, the sample was dialyzed against an EDTA solution to remove Cu ions and to avoid possible toxicity caused by this heavy metal, as mentioned before [43,44]. The conjugation efficiency was rather high, 80.4%, as reported before by Hein and Fokin [45]. The synthesized DOX-propGA3-polymer conjugate contained 7 wt% DOX.

The DOX-propGA3-polymer conjugate was further characterized using GPC with dual UV (480 nm to detect DOX) and RI detection (Figure 2). The chromatogram of the physical mixture of p(HEMAm-co-AzEMAm)-Gly-HEMAm and DOX-propGA3 displayed a RI peak of the polymer with retention time of 12.4 min and a UV peak of the prodrug with retention time of 16.5 min (Figure 2A). After the click chemistry reaction, the obtained product eluted at 12.4 min (RI detection) and the UV peak shifted from 16.5 min to 12.4 min (Figure 2B), which demonstrates that DOX was indeed successfully conjugated to the polymer. A small peak was observed at the retention time of free DOX-propGA3. Calculation of the area under the curve of conjugated and free DOX-propGA3 prodrug shows that there was approximately 5% of free prodrug DOX-propGA3 present in the final product. This trace amount of free propGA3-DOX was most likely washed away during the nanogel preparation procedure and therefore not present in the final formulation added to the cells.

The DOX-propGA3-polymer was subsequently used in the preparation of nanogels via mini-emulsion photopolymerization, Figure 1B. The methacrylamide groups at the side chain of the conjugate were crosslinked under UV. The size and ζ-potential of DOX-propGA3-nanogels were 164 nm (PDI 0.14) and −2.7 ± 0.1 mV, respectively, which was similar to empty nanogels (size 172 nm, PDI 0.16 and ζ-potential −2.5 ± 0.2 mV). This indicated that conjugation of DOX-propGA3 to the polymer did not affect the size and ζ-potential of the formed nanogels.

### 3.2. Conversion of Prodrug into DOX by Bovine β-Gus

Figure 3A shows the percentage converted DOX from the DOX-propGA3-polymers and DOX-propGA3-nanogels in the presence and absence of bovine β-gus over time, in PBS supplemented with 0.1% BSA. Only in the presence of bovine β-gus were the DOX-propGA3-polymer and DOX-propGA3-nanogel converted into DOX. This indicates that there was no chemical conversion of DOX-propGA3 into DOX, which is in line with other prodrugs with similar structures [10,15]. Complete conversion of DOX-propGA3-polymer and DOX-propGA3-nanogel into DOX was obtained after 24 and 48 h, respectively. The complete conversion of DOX-propGA3-nanogel into DOX was slower than DOX-propGA3-polymer. This difference was most likely related to the difference in structure between DOX-propGA3-polymer and DOX-propGA3-nanogels. Since β-gus has a rather high molecular weight (>300 kDa) [46], the β-gus is not able to enter the nanogels. Therefore, the DOX-GA3 (a substrate of β-gus) first needs to be released from the nanogels. The release of DOX-GA3 from the nanogel most likely takes place by either diffusion of DOX-GA3 out of the nanogels after hydrolysis of the ester between the triazole and DOX-GA3 or by nanogel degradation leading to (free) polymer chains, Figure 3B. Subsequently, the β-gus is able to convert the DOX-GA3 into DOX, whereas for the DOX-propGA3-polymer only the ester group between the triazole and prodrug needs to be hydrolyzed before DOX-GA3 is released, that is, subsequently quickly converted into DOX by β-gus. As a consequence, DOX formation from the nanogels was slower than from the polymer conjugate. In contrast, the conversion rate of DOX-propGA3-nanogel into DOX is much faster than previously designed micelles containing DOX-propGA3, viz. 100% conversion after 2 days incubation compared to 25–35% conversion after 4 days incubation [32,39], respectively. This could be due to the lower water activity in the hydrophobic core of micelles, compared to the nanogels, resulting in a slower hydrolysis of the ester bond connecting the prodrug and the polymer backbone.

### 3.3. Exposure of 4T1 Cells to HIFU and Conversion of Prodrug by HIFU Treated Cells

Figure 4 shows the microscopic images of untreated cells (A) and cells exposed to HIFU (B) at a magnification of 10×. Untreated cells were round and had a smooth surface, representing normal physiology of the cells 1 h after plating. In the sample of cells exposed to HIFU, only cell debris was present and no viable cells were observed. The supernatant of cells exposed to HIFU contained a β-gus enzyme activity of 7.3 ± 0.7 units/1 × 10^6^ cells, whereas the supernatant of untreated cells contained a β-gus enzyme activity of only 0.31 ± 0.1 units/1 × 10^6^ cells. The supernatant of cells exposed to HIFU was able to convert the DOX-propGA3-nanogels completely into DOX within 48 h. These results were in line with the DOX release results from DOX-propGA3-nanogels with bovine β-gus.

### 3.4. In Vitro Cytotoxicity

Figure 5 shows cell viability of cells treated with different concentrations of DOX, DOX-propGA3, DOX-propGA3-polymer, and DOX-propGA3-nanogels, in complete cell culture medium, supplemented with (A) PBS (negative control), (B) bovine β-gus, and (C) supernatant of 4T1 cells exposed to HIFU. Treatment of cells with DOX-propGA3, DOX-propGA3-polymer, and DOX-propGA3-nanogels in complete cell culture medium supplemented with 5% PBS (Figure 5A) did not result in cytotoxicity, except for cells treated with DOX-propGA3-nanogels at a concentration of 1 mM, the highest concentration investigated. These results are in line with previous studies with comparable prodrugs [15,39]. Cells treated with DOX-propGA3, DOX-propGA3-polymer, and DOX-propGA3-nanogel, in combination with bovine β-gus at a concentration of 50 µg/mL (Figure 5B) and supernatant of cells exposed to HIFU (Figure 5C), experienced at an increase in concentration, a decrease in cell viability. Both bovine β-gus and supernatant of cells exposed to HIFU significantly increased the cytotoxicity of the different prodrug formulations to a similar extent. In all conditions, cells treated with DOX showed the largest cytotoxicity (IC_50_ of 2,000 nM), Table 1. Lysate of cells exposed to HIFU caused limited cytotoxicity, and cell viability of 93.6 ± 3.9%. In addition, empty nanogels have good cytocompatibility at the used concentrations [40]. This indicates that the cytotoxicity of the nanogels in combination with liberated β-gus from cells exposed to HIFU was caused by the converted prodrug, released from the nanogels. The cytotoxicity of DOX was not influenced by the β-gus or supernatant of cells exposed to HIFU. Therefore, DOX-propGA3-nanogel is a promising formulation since it only converts into DOX in the presence of β-gus and it does not result in cytotoxicity in the absence of this enzyme.

It has been observed before that nanogels can be internalized by cells by endocytosis and end in endosomes and lysosomes [25]. These lysosomes contain the β-gus enzyme [47]. Therefore, specific activation of the prodrug into the cytotoxic drug could occur. However, the pH in these lysosomes is rather low (pH between 4.5 and 5 [48]). Since the hydrolysis rate of ester bonds is at a low pH [49], the hydrolysis of DOX-propGA3 into DOX-GA3 is hampered in these lysosomes. Therefore, DOX-propGA3-nanogels will not cause cytotoxicity in the normal cells when DOX-propGA3-nanogels are endocytosed in these cells.

These results motivate further in vitro testing of this proof of principle. In vitro experiments are required to optimize the tumor volume that is exposed to HIFU in order to liberate their β-gus for prodrug conversion, released from nanogels, into the chemotherapeutic agent doxorubicin in order to kill the remaining tumor cells in the tumor margin.

## 4. Conclusions

A DOX-glucuronide prodrug (DOX-propGA3) was conjugated to the polymer p(HEMAm-co-AzEMAm)-Gly-HEMAm by click chemistry (to yield DOX-propGA3-polymer). Subsequently, this DOX-propGA3-polymer was used to prepare DOX-propGA3-nanogels. The glucuronide spacer was selectively cleaved by liberated β-gus from cells exposed to HIFU. Furthermore, the supernatant of cells exposed to HIFU increased the cytotoxicity of DOX-propGA3-polymer and DOX-propGA3-nanogels due to liberated β-gus from 4T1 cells. Therefore, DOX-propGA3-nanogels in combination with HIFU treatment of the tumor could be a novel and attractive therapeutic modality for anticancer therapy.

## Figures and Tables

**Figure 1 pharmaceutics-12-00536-f001:**
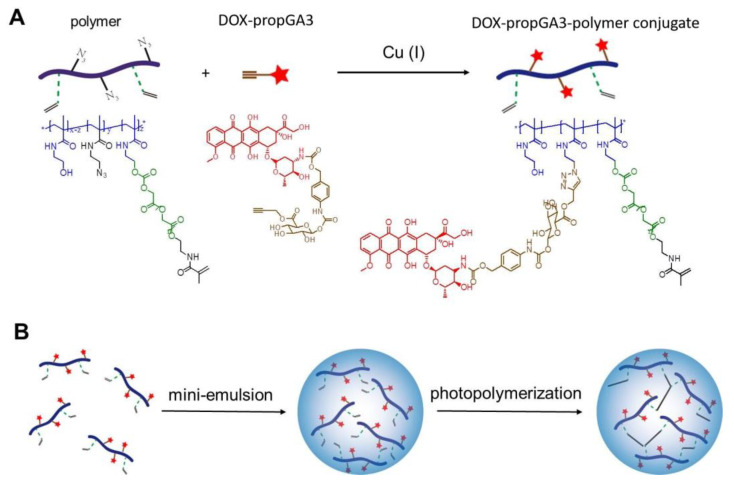
Schematic representation of the synthesis of the doxorubicin-glucuronide prodrug (DOX-propGA3)-polymer and DOX-propGA3-nanogels. (**A**) Synthesis of DOX-propGA3-polymer conjugate using click-chemistry and (**B**) preparation of prodrug-loaded nanogels from DOX-propGA3-polymer conjugates using inverse mini-emulsion photopolymerization.

**Figure 2 pharmaceutics-12-00536-f002:**
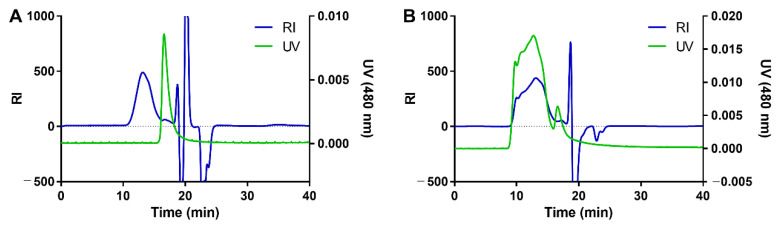
GPC analysis with dual refractive index (RI) and ultraviolet (UV) (480 nm) detection of (**A**) physical mixture of p(HEMAm-co-AzEMAm)-Gly-HEMAm and DOX-propGA3, and (**B**) DOX-propGa3-polymer conjugate.

**Figure 3 pharmaceutics-12-00536-f003:**
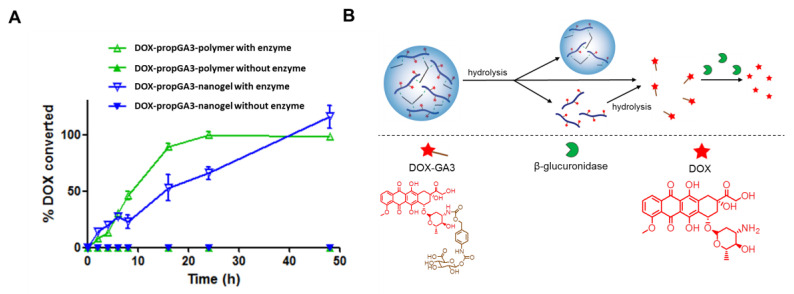
Conversion of DOX-propGA3-nanogels and DOX-propGA3-polymer into DOX. (**A**) Conversion profile of DOX from DOX-propGA3-polymer and DOX-propGA3-nanogels with or without bovine β-gus at a concentration of 100 units/mL (*n* = 3). (**B**) Schematic representation of prodrug conversion form the nanogel into DOX.

**Figure 4 pharmaceutics-12-00536-f004:**
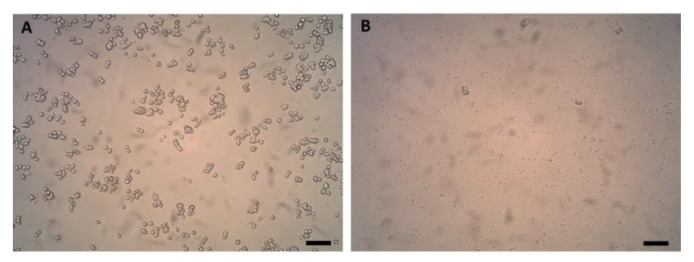
Bright field microscopy images with a magnification of 10× of (**A**) untreated cells and (**B**) cells after exposure to HIFU for 10 min with a peak negative pressure of 41 MPa; bar represents 500 µm.

**Figure 5 pharmaceutics-12-00536-f005:**
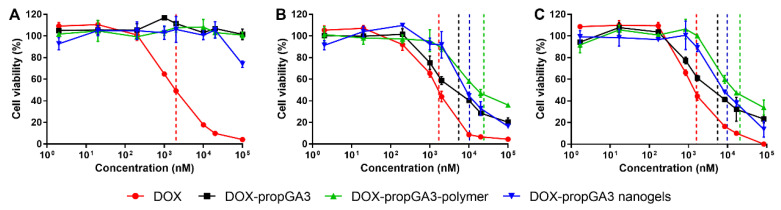
Viability of 4T1 cells incubated with doxorubicin (DOX), DOX-propGA3, DOX-propGA3-polymer, and DOX-propGA3-nanogels with PBS (**A**), bovine β-gus (**B**), and supernatant of HIFU-treated cells (**C**) cells (*n* = 3). Dashed lines represent the IC_50_ of each treatment.

**Table 1 pharmaceutics-12-00536-t001:** IC_50_ (nM) of 4T1 cells incubated with DOX, DOX-propGA3, DOX-propGA3-polymer, and DOX-propGA3-nanogels, with PBS, bovine β-gus, and supernatant of cells, exposed to HIFU. * *p* < 0.05 between PBS and bovine β-gus or supernatant of cells exposed to HIFU.

	IC_50_ with PBS (nM)	IC_50_ with Bovine β-Gus (nM)	IC_50_ with Supernatant of Cells Exposed to HIFU (nM)
DOX	2000 ± 300	1700 ± 200	1600 ± 300
DOX-propGA3	>100,000	5500 ± 1100 *	5600 ± 1400 *
DOX-propGA3-polymer	>100,000	24,100 ± 4700 *	2100 ± 1800 *
DOX-propGA3 nanogels	>100,000	10,300 ± 1800 *	9900 ± 1100 *

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
