# Peer review of "A Doxorubicin-Glucuronide Prodrug Released from Nanogels Activated by High-Intensity Focused Ultrasound Liberated β-Glucuronidase"

_pharmaceutics, 2020, doi:10.3390/pharmaceutics12060536_

Round 1

Reviewer 1 Report

Authors present studies on application of new type of nanogel preparation for topical treatment that contains doxorubicine in the pro-drug form covalently linked to the polymeric material. The drug is efficienly released from the polymeric material due to high-intensity focused ultrasound (HIFU) via release of exoogenous  β-glucuronidase present preferably in malignant cells. Authors confirmed in in vitro studies that the same effect was achieved if bowine β-glucuronidase was added to the preparation.

My only concern is focused on application of high-intensity focused ultrasound (HIFU) conditions that might be dangerous for the skin and induce both healthy and malignant cells destruction and leak genetic material of the malignant cells to the neighboring tissues what may result with metastasis. It is fine with single application. In case of multiple application therapeutic effect might be overcomed by unwanted side effect. Since no in vivo studies on long term application of HIFU conditions is presented, this method seem to be limited to several uses. Please comment on that

Author Response

First of all we would like to thank the reviewers for their constructive comments that helped us to improve the manuscript.

Reviewer 1

Authors present studies on application of new type of nanogel preparation for topical treatment that contains doxorubicine in the pro-drug form covalently linked to the polymeric material. The drug is efficienly released from the polymeric material due to high-intensity focused ultrasound (HIFU) via release of exoogenous β-glucuronidase present preferably in malignant cells. Authors confirmed in in vitro studies that the same effect was achieved if bowine β-glucuronidase was added to the preparation.

Question 1: My only concern is focused on application of high-intensity focused ultrasound (HIFU) conditions that might be dangerous for the skin and induce both healthy and malignant cells destruction and leak genetic material of the malignant cells to the neighboring tissues what may result with metastasis. It is fine with single application. In case of multiple application therapeutic effect might be overcomed by unwanted side effect. Since no in vivo studies on long term application of HIFU conditions is presented, this method seem to be limited to several uses. Please comment on that

Response: High acoustic pressures are achieved in the focus of the ultrasound beam, resulting in destruction of the tumor cells only in the focus. At locations outside the focus the acoustic pressure is significantly lower, therefore damage to tissues located outside the acoustic focus, like the skin, is minimized (Ter Haar and Coussios, International Journal of hyperthermia, 2007).

According to the literature mechanical HIFU will not lead to extra metastases, see Schade et al, Journal of Urology, in 2012 and Xing et al, Biochemical and Biophysical Research Communications, in 2009. In contrast, there is increasing proof that mechanical HIFU can induce an immune response in the treated tumor leading to a systemic effect (Van den Bijgaart et al, Cancer Immunology, Immunotherapy, in 2016 and Khokhlova et al, International journal of hyperthermia, in 2015).

Changes in the text: Added to line 87 “, without damaging the normal tissue”

Reviewer 2 Report

The article showed an interesting approach to use HIFU as a remote control power output to triggered enzyme therapy by the release of doxorubicin from the developed nanogels. Although the concept is very interesting, the therapeutic efficiency of the nanogels developed is only showed when nanogels are incubated in PBS with the target cells. These assays has to be done in complete cell culture media to be able to evaluate the potential use of the designed nanogels in the therapeutic scheme that is proposed. The following suggestions could help authors to include relevant information on the article that I think is necessary for its publication.

Major comments:

1)- All the assays done with cells are carried out in PBS. What happens if they are carried out in complete cell media?. A big issue of nanotherapeutics is their stability in complex media as their aggregation could hinder their use. Are the nanogels stable in complete media? What about their therapeutic efficiency in complete cell culture media? If the nanogels are not colloidal stable their composition should be modified using blocking agents (eg: PEGs) that could enhance their stability due to a steric effect. As an enzyme triggers the therapeutic action of these nanogels, the blocking of the nanogel to improve their colloidal stability could affect the release of the drug by the enzyme due to also steric issues. Thus, this is not a minor point to be optimized to assess the real potential of the nanogels in the proposed therapy scheme. Authors did not mention this issue and I think they should clarify this aspect as it could happen that the current design of the proposed nanogels could not work in the conditions usually used for studying therapeutic efficiency in vitro. Indeed, all cytotoxicity studies carried out in their previous paper (reference 38) were conducted in complete cell culture media, the PEGylation of the nanogels has been conducted and the stability of the nanogels on cell culture media was also previously study.

2)- In the cytotoxicity studies showed in Figure 5A, it is clear that the cytotoxicity is triggered by the enzyme in the case of the nanogels and the intermediates used for their preparation. However as these experiments are carried out in PBS for 24h, it should be interesting that authors mention the effect on cell viability of this incubation condition with respect to the incubation for the same time of period but in complete cell media. The cells in PBS are stressed if long incubation times are used, thus the cytotoxic effect of the therapeutics could be very different compared to non-stressed cells. The dose to be used of nanogels could be higher in the case of non-stressed cells and cytotoxicity of the nanogels (without adding the enzyme) is observed at the highest concentrations used for these experiments. It is true that authors had explained that this should be  a consequence of the internalization of the nanogels and the release of the drug due to the pH and/or the presence of enzymes on lysosomes. But from the data showed, an inherent cytotoxicity of the nanogels could not be ruled out. Thus, as empty nanogels have similar size and zeta-potential to DOX-loaded ones, it should be interesting to show the cytotoxicity of empty nanogels at least in PBS without the addition of beta-glucoronidase.

3)- Another important control that is missing in Fig 5 experiments is to show the cytotoxic effect of the supernatant of cells exposed to HIFU. It is well known that lysosome disruption could trigger cell death itself, so which is the cytotoxicity baseline caused by the incubation of the target cells with the supernatant obtained after triggering lysosomes disruption by HIFU treatment?. This is also an important clarification to be able to evaluate the therapeutic potential of the developed nanogels.

4)- With respect to HIFU as remote control for triggering the release of endogenous b-glucoronidase and thus the release of DOX from the nanogels, it should be also interesting that authors include in the introduction a clarification of how this technique is implemented to avoid cytotoxicity due to ablation of the tumour tissue and/or cell death due to the release of lysosome content. This is not the objective of the use of HIFU on the proposed therapeutic approach, and thus information on how this could be controlled to achieve a mild HIFU treatment should be clarified. The conditions used of HIFU to obtain endogenous b-gluc with enough concentration to trigger the release of DOX from the nanogels were clearly cytotoxic for the cells that were exposed to this treatment (Figure 4). Is it possible to achieve a release of lysosome contents enough to ensure the bioconversion of the gels but that does not to trigger cell death of the HIFU treated cells?.

Minor comments:

1)- As the preparation of the DOXpropGA3-polymer is carried out by click chemistry, I think that authors should give more details on how the dialysis is carried out (time, number of changes, volume ratio between sample and volume of solvent). Is there any quality control established about the remaining amount of cooper, eg ICP analysis of each batch of nanogels.

2)- Authors should clarify if DLS measurements are expressed on intensity or number. 

Author Response

First of all we would like to thank the reviewers for their constructive comments that helped us to improve the manuscript.

Reviewer 2

The article showed an interesting approach to use HIFU as a remote control power output to triggered enzyme therapy by the release of doxorubicin from the developed nanogels. Although the concept is very interesting, the therapeutic efficiency of the nanogels developed is only showed when nanogels are incubated in PBS with the target cells. These assays has to be done in complete cell culture media to be able to evaluate the potential use of the designed nanogels in the therapeutic scheme that is proposed. The following suggestions could help authors to include relevant information on the article that I think is necessary for its publication.

Major comments:

1)- All the assays done with cells are carried out in PBS. What happens if they are carried out in complete cell media?. A big issue of nanotherapeutics is their stability in complex media as their aggregation could hinder their use. Are the nanogels stable in complete media? What about their therapeutic efficiency in complete cell culture media? If the nanogels are not colloidal stable their composition should be modified using blocking agents (eg: PEGs) that could enhance their stability due to a steric effect. As an enzyme triggers the therapeutic action of these nanogels, the blocking of the nanogel to improve their colloidal stability could affect the release of the drug by the enzyme due to also steric issues. Thus, this is not a minor point to be optimized to assess the real potential of the nanogels in the proposed therapy scheme. Authors did not mention this issue and I think they should clarify this aspect as it could happen that the current design of the proposed nanogels could not work in the conditions usually used for studying therapeutic efficiency in vitro. Indeed, all cytotoxicity studies carried out in their previous paper (reference 38) were conducted in complete cell culture media, the PEGylation of the nanogels has been conducted and the stability of the nanogels on cell culture media was also previously study.

Response:

The reviewers is obviously put on the wrong track by the used media; therefore we clarified it more clearly in the revised manuscript. The conversion of the nanogels into DOX with bovine β-gus and supernatant of cells exposed to HIFU were performed in PBS with 0.1% BSA. The cytotoxicity assays were performed in complete cell culture medium (RPMI 1640 + 10% FBS) supplemented with 5% PBS.

Changes in the text: Removed from line 320 “in combination with” and line 323 “dispersed in” Added to line 221-226 “cell culture medium”, “line 277 “, in PBS supplemented with 0.1% BSA” and line 320 “in complete cell culture medium, supplemented with” and line 323 “in complete cell culture medium supplemented with 5% PBS”

2)- In the cytotoxicity studies showed in Figure 5A, it is clear that the cytotoxicity is triggered by the enzyme in the case of the nanogels and the intermediates used for their preparation. However as these experiments are carried out in PBS for 24h, it should be interesting that authors mention the effect on cell viability of this incubation condition with respect to the incubation for the same time of period but in complete cell media. The cells in PBS are stressed if long incubation times are used, thus the cytotoxic effect of the therapeutics could be very different compared to non-stressed cells. The dose to be used of nanogels could be higher in the case of non-stressed cells and cytotoxicity of the nanogels (without adding the enzyme) is observed at the highest concentrations used for these experiments. It is true that authors had explained that this should be  a consequence of the internalization of the nanogels and the release of the drug due to the pH and/or the presence of enzymes on lysosomes. But from the data showed, an inherent cytotoxicity of the nanogels could not be ruled out. Thus, as empty nanogels have similar size and zeta-potential to DOX-loaded ones, it should be interesting to show the cytotoxicity of empty nanogels at least in PBS without the addition of β-glucoronidase.

Response: We agree with the reviewer that incubation of cells in 100% PBS for 24 hours causes stress to the cells and could affect the cytotoxicity of other compounds. However, for the cytotoxicity assays we incubated the cells in complete cell culture media supplemented with 5% PBS (discussed above and described in the Materials and Methods section on page 6 of the manuscript).

Empty nanogels have a good cytocompatibility at the concentrations used in this study, as observed before by Chen et al, Macromolecular Bioscience, in 2016 (Polymeric Nanogels with Tailorable Degradation Behavior).

Changes in the text: Removed from line 320 “in combination with” and line 323 “dispersed in” Added to line 221-226 “cell culture medium”, “line 320 “in complete cell culture medium, supplemented with”, line 323 “in complete cell culture medium supplemented with 5%” and line 333 “In addition, empty nanogels have a good cytocompatibility at the used concentrations [40].”

3)- Another important control that is missing in Fig 5 experiments is to show the cytotoxic effect of the supernatant of cells exposed to HIFU. It is well known that lysosome disruption could trigger cell death itself, so which is the cytotoxicity baseline caused by the incubation of the target cells with the supernatant obtained after triggering lysosomes disruption by HIFU treatment?. This is also an important clarification to be able to evaluate the therapeutic potential of the developed nanogels.

Response: We agree with the reviewer that the supernatant of cells exposed to HIFU may affect the cell survival of cells that are not exposed to HIFU. This control experiment was indeed performed, but we forgot to put the data in the first version of the manuscript. Incubating pre-seeded 4T1 cells with supernatant of 4T1 cells after exposing them to HIFU resulted in a cell viability of 93.6±3.9%, indicating that lysate of cells exposed to HIFU had limited effect on the cytotoxicity of the untreated cells. We included this in the reversed manuscript.

Changes in the text: Added to line 332 “Lysate of cells exposed to HIFU caused limited cytotoxicity, cell viability of 93.6±3.9%.” and line 334 “This indicates that the cytotoxicity of the nanogels in combination with liberated β-gus from cells exposed to HIFU was caused by the converted prodrug, released from the nanogels.”

4)- With respect to HIFU as remote control for triggering the release of endogenous b-glucoronidase and thus the release of DOX from the nanogels, it should be also interesting that authors include in the introduction a clarification of how this technique is implemented to avoid cytotoxicity due to ablation of the tumour tissue and/or cell death due to the release of lysosome content. This is not the objective of the use of HIFU on the proposed therapeutic approach, and thus information on how this could be controlled to achieve a mild HIFU treatment should be clarified. The conditions used of HIFU to obtain endogenous b-gluc with enough concentration to trigger the release of DOX from the nanogels were clearly cytotoxic for the cells that were exposed to this treatment (Figure 4). Is it possible to achieve a release of lysosome contents enough to ensure the bioconversion of the gels but that does not to trigger cell death of the HIFU treated cells?.

Response: For in vivo experiments we aim to destroy the tumor bulk with mechanical HIFU and as a consequence liberate β-gus. Subsequently, the liberated β-gus is able to convert the prodrug, released from the nanogels, into the chemotherapeutic agent that will kill the remaining tumor cells in the tumor margin.

Changes in the text: Added to line 352 “These results motivate further in vivo testing of this proof of principle. In vivo experiments are required to optimize the tumor volume that is exposed to HIFU in order to liberate their β-gus for prodrug conversion, released from nanogels, into the chemotherapeutic agent doxorubicin in order to kill the remaining tumor cells in the tumor margin.”

Minor comments:

5)- As the preparation of the DOXpropGA3-polymer is carried out by click chemistry, I think that authors should give more details on how the dialysis is carried out (time, number of changes, volume ratio between sample and volume of solvent). Is there any quality control established about the remaining amount of cooper, eg ICP analysis of each batch of nanogels.

Response: As mentioned in the manuscript at line 131, 'dialyzed (membrane cut-off 3.5 kDa) against ammonium acetate buffer (20 mM, pH 5, containing 10 mM EDTA) for 2 days, followed by dialysis against water for 24 hours'. The dialysis buffer/water was changed at least 6 times. The volume ratio between dialysis buffer/waster and sample was larger than 500.

We did not perform a quality control on the amount of remaining copper. Removal of copper by the dialysis against a buffer containing EDTA is proven by others, [Hein et al. Click Chemistry, A Powerful Tool for Pharmaceutical Sciences, Pharmaceutical Research, 2008] and [Presolski et al. Copper-Catalyzed Azide–Alkyne Click Chemistry for Bioconjugation, Curr Protoc Chem Biol. 2011]

Changes in the text: Added to line 133: “The ammonium acetate buffer and water were changed at least 6 times. Ratios between ammonium acetate buffer and sample and between water and sample were larger than 500.” Added to line 251 “as mentioned before [43,44].”

6)- Authors should clarify if DLS measurements are expressed on intensity or number.

Response: The DLS measurements are expressed on intensity

Changes in the text: Added to line 162 “expressed on intensity”

Round 2

Reviewer 2 Report

Authors had clarified all te points I have mentioned in my first review. They have introduced theses clarification in the text and even missing control experiments. Thus, I think that now the article could be published in its present form.